# Optimizing Face Recognition Inference with a Collaborative Edge–Cloud Network

**DOI:** 10.3390/s22218371

**Published:** 2022-11-01

**Authors:** Paul P. Oroceo, Jeong-In Kim, Ej Miguel Francisco Caliwag, Sang-Ho Kim, Wansu Lim

**Affiliations:** 1Department of Aeronautics, Mechanical and Electronic Convergence Engineering, Kumoh National Institute of Technology, Gumi 39177, Korea; 2Department of Industrial Engineering, Kumoh National Institute of Technology, Gumi 39177, Korea

**Keywords:** deep learning, edge–cloud, face recognition, real-time, TCP/IP

## Abstract

The rapid development of deep-learning-based edge artificial intelligence applications and their data-driven nature has led to several research issues. One key issue is the collaboration of the edge and cloud to optimize such applications by increasing inference speed and reducing latency. Some researchers have focused on simulations that verify that a collaborative edge–cloud network would be optimal, but the real-world implementation is not considered. Most researchers focus on the accuracy of the detection and recognition algorithm but not the inference speed in actual deployment. Others have implemented such networks with minimal pressure on the cloud node, thus defeating the purpose of an edge–cloud collaboration. In this study, we propose a method to increase inference speed and reduce latency by implementing a real-time face recognition system in which all face detection tasks are handled on the edge device and by forwarding cropped face images that are significantly smaller than the whole video frame, while face recognition tasks are processed at the cloud. In this system, both devices communicate using the TCP/IP protocol of wireless communication. Our experiment is executed using a Jetson Nano GPU board and a PC as the cloud. This framework is studied in terms of the frame-per-second (FPS) rate. We further compare our framework using two scenarios in which face detection and recognition tasks are deployed on the (1) edge and (2) cloud. The experimental results show that combining the edge and cloud is an effective way to accelerate the inferencing process because the maximum FPS achieved by the edge–cloud deployment was 1.91× more than the cloud deployment and 8.5× more than the edge deployment.

## 1. Introduction

With the development of deep learning (DL) applications, the implementation of real-time video and image analytics in computer vision (CV) has become an active field of research in recent years. The applications of CV proved to be a great problem solver in real-time applications. Several CV application studies have been carried out, such as detecting a compact fluorescence lamp [1], recognizing speed limit traffic signs using a shape-based approach [2], detecting sugar beetroot crops with mechanical damage [3], designing of an autonomous underwater vehicle that performs computer-vision-driven intervention tasks [4], tracking of ball movement in a smart goalkeeper prototype [5], and recognizing an obstacle on a powered prosthetic leg [6]. These papers all utilized a single-board computer for real-time deployment. The authors of [7] discussed the recent developments in the computer vision domain, particularly in face detection sector, focusing on the development of 2D facial recognition to optimize the recent systems that use 3D face recognition algorithms [8]. The authors of [9] proposed a recognition system using ear images based on an unsupervised deep learning technique. Cameras generate a huge amount of sensor data, which needs to be processed for video analytics applications. These applications can be applied in several fields, including thermal camera applications [10], autonomous vehicles [11], traffic engineering and monitoring, access control (e.g., buildings, airports), logistics, and biometrics [12]. This means that DL-based computer vision applications for video analytics are anticipated to yield a huge amount of data. Consequently, the real-time demands of these applications lead to several challenges in terms of storage, communication, and processing [13].

One common application of video analytics is face recognition. DL-based face recognition has two parts: training and inference. It is more suitable to perform the training process in the cloud because the process requires high throughput computing and a huge amount of data [14,15]. The cloud can provide sufficient resources for processing, storage, and energy. However, it does not scale well as the number of cameras increases. Thus, the inference process, which requires low latency, cannot be handled by cloud computing [16]. However, while the edge is more flexible and can scale well as the number of cameras increases, it is inadequate with regards to processing, storage, and energy [15]. Numerous studies have explored the dilemma of having to choose between these two strategies and proposed various approaches. Most of these approaches focused mainly on independently improving the performance of DL-based applications either on the edge or the cloud. However, it is imperative to explore a solution that tries to concurrently improve the performance of these applications on both the edge and cloud [17]. One way to achieve this is to explore a solution that has a tradeoff between managing the limited computational capabilities of the edge device and latency issues in the cloud. The integration of edge and cloud for computing purposes can lead to a hierarchical mode of computing in which tasks are processed by both the edge node and cloud in an opportunistic fashion [18,19].

Recently, some researchers have attempted to optimize deep learning applications by combining the edge and cloud. In [20], the authors proposed three hierarchical optimization frameworks that aim to reduce energy consumption and latency. Their strategy was to offload tasks strategically from the edge node to the cloud node. In [21], an offloading framework based on edge learning was proposed for autonomous driving. Although these studies reinforce our claim that the edge and cloud should be combined for optimal performance, they are based entirely on simulations because implementations with actual devices have not been considered. In [22], a real-time baby facial expression recognition system was proposed. In this system, the predictions are stored on the actual edge device, whereas insights are sent to the cloud at intervals for visualization and analysis. Although this work efficiently implements and tries to combine the edge and cloud for optimized learning, (i) it only allocates noncomputationally intensive tasks to the cloud and (ii) it performs most of the training, detection, and recognition on the edge device. The authors of [23] proposed a system that automatically partitioned computations among several cloud servers, proving that this method provides faster inference speed; however, they focused on utilizing multiple cloud servers rather than mainly collaborating with the edge device. In [24], an optimized edge implementation of face recognition is proposed but fails to utilize the capability of cloud as a processing tool and is only used as storage. This not only defeats the purpose of a collaborative edge–cloud network but also leads to computational pressure on the edge device or on the cloud server and a suboptimal learning process. Moreover, [25] noted that transferring all tasks from the edge to the cloud could lead to second-level latency, which does not meet the real-time requirement.

In this study, we provided an implementation of optimized inference of face recognition utilizing a collaborative edge–cloud network. The main contribution of this paper is as follows: (i) we implemented an edge–cloud-based real-time face recognition framework based on the multitask cascaded convolutional neural network (MTCNN) consisting of P-net, R-net, and O-net and trained it using randomly cropped patches from WIDER FACE dataset for positives, negatives, and part face with additional data from cropped faces from the CelebA dataset as landmark faces [26] and the Python face recognition library, Dlib-ml [27]. In the proposed framework, once the on-device face detection acquired a face extracted from video frames, the system will then send the face image instead of the whole frame to the cloud server, resulting in significantly faster inference, and communication will use less bandwidth and energy. The server-side system will then perform face recognition. (ii) We implemented communication between the edge and cloud using the TCP/IP of wireless communication. We set up a TCP server on the edge, which sends video frames to the TCP client on the cloud for onward processing. (iii) Furthermore, we compared the proposed system with two scenarios: (a) Performing both the detection and recognition tasks at the edge device; and (b) performing both the detection and recognition tasks in the cloud.

## 2. Review of Related Works

Several approaches have been proposed for the real-time execution of face recognition tasks, based on the deployment architecture. The ultimate aim is to identify the best way to connect cameras to computing devices such as the edge and cloud while satisfying the latency and energy-efficiency requirements. The goal is to achieve the highest frame per second (FPS) rate and concurrently maintain a low energy consumption level. A typical face recognition task consists of two subtasks, which are independently computationally extensive. These could lead to increased energy consumption and a surge in the processing time. These tasks include: (i) face detection and (ii) face recognition, which are both based on deep learning. The major challenge is the decision to either deploy the architecture on the cloud or on the edge. This section discusses the two possible deployments and their disadvantages.

### 2.1. Face Recognition

Face recognition is the capability of human beings to differentiate one face from another. This concept is adapted in machines utilizing deep learning techniques, producing models that can recognize faces in videos or images. The research on machine recognition of faces began in the year 1970 [28]. During the past decades, the development of face recognition algorithms have gained attention due to the abundant data available [29]. In [30], a support vector machine (SVM) is used for classification problems such as face recognition. An SVM is a pattern classifier that considers form as a matching scheme in controlled and uncontrolled situations. An artificial neural network is developed for face recognition tasks. In [30] a feed-forward algorithm, a multilayer perceptron (MLP) is proposed for supervised pattern matching and has been used in several pattern classification tasks such as face recognition. Another face detection algorithm is the Gabor wavelets with feed-forward algorithm. This method is used for finding feature points and extracting feature vectors. A new type of convolutional neural network is proposed in [31]. In this method, the inhibitory neurons are shunned by the processing cells. In the past, it has been demonstrated that shunning inhibitory neurons are more effective than MLPs for classification and nonlinear regression when utilized in a traditional feed-forward architecture. They are substantially better than MLPs at approximating complex decision surfaces. Most of these methods are computationally expensive and complex tasks, thus requiring an ample amount of processing power and high memory for training and inferencing.

### 2.2. Cloud-Based Machine Learning

Cloud-based deployment refers to the processing of face recognition tasks on the cloud. It is also termed as computation offloading [20]. This is because in the literature, tasks that are computationally intensive are offloaded from the edge to the cloud. In a typical cloud-based deployment, the camera is connected to a cloud server via an RTSP stream over the internet or using a local network. Owing to the distance of the sensor (i.e., camera) from the cloud, the frame rate is affected by the available bandwidth in the communication channel. Videos are streamed from the camera to the cloud. The cloud receives the camera frames at a certain FPS and directs the frame to the DL model, which executes both tasks of face detection and recognition.

Like the edge, the biggest challenge in cloud-based face recognition is whether the cloud capabilities are sufficient to process the incoming frames in real time and with a fair FPS rate. One major problem is that the FPS might be considerably low because of the constraints of the communication channel. However, owing to the high processing performance of the GPUs of cloud servers, the processing of DL frameworks will be seamless. Thus, in this deployment, the performance of the face recognition process is significantly affected by the communication channel when receiving the continuous video stream from the camera. As the number of cameras increases, it becomes less scalable. Furthermore, there are also security and privacy concerns with the deployment of crucial data to the cloud.

### 2.3. Edge-Based Machine Learning

Edge-based deployment of machine learning models refers to embedded machine learning. Embedded machine learning is a concept that has been gaining popularity due to the development of several ways of implementing machine learning models in power-constrained devices such as efficient neural networks, optimization techniques, and edge-oriented frameworks. In [32,33], they established an edge computing paradigm in video surveillance units (VSUs) by locally processing captured images. In [34], a method is proposed where a computer vision system is implemented on an edge device that combines a gray level co-occurrence matrix and a support vector machine (SVM) that enables the system to be implemented on a minimum resource platform. Another computer vision system uses SVM to classify obstacles, which is implemented in a single-board computer (SBC), which is capable of running less complex ML models. They also utilized the cloud for database storage [35]. Edge devices that implement computer vision such as obstacle detection and face recognition are devices that are usually embedded GPU computing devices, such as a Jetson board. In this deployment, the edge device consists of the camera and an embedded device, such as a GPU computing device. Videos are streamed from the camera to the computing device. The computing device receives the camera frames at a certain FPS and performs the dual operation of face detection and face recognition using deep learning algorithms. The concept behind this deployment is that the computation tasks are executed close to the data source (i.e., cameras).

The biggest challenge in edge-based deployment is whether the capabilities of the edge device are adequate to process the incoming frames in real time and with a fair FPS rate. The solution lies in the specification of the computing device and DL framework employed. Computing devices such as the Jetson Nano have less storage and computing resources when compared with the Jetson Xavier AGX. Similarly, the inference performance of the face recognition process significantly depends on the selected deep learning framework. The most common frameworks include TFLite from Google and TensorRT from NVIDIA. TensorRT is preferred because it is capable of speeding up the inference performance of deep learning models running on Jetson Nano boards. This is because it can maximize the number of inference images processed by quantization. It also optimizes the bandwidth and memory usage of the computing device by performing layer and tension fusion. Thus, in this deployment, the performance of the face recognition process significantly depends on the device and DL framework being used. In terms of security and privacy, this deployment is adequate, as it has total control over what data is being received or sent.

### 2.4. Communication Protocol

There are various types of existing communication protocols that are used in different applications. The following exist for a normal communication protocol: (1) a user datagram protocol (UDP), which is a substitute communication protocol to transmission control protocol implemented primarily for creating loss-tolerating and low-latency linking between different applications; (2) post office protocol (POP), which is designed and used for receiving e-mails; and (3) simple mail transport protocol (SMTP), designed to send and distribute outgoing e-mail. Moreover, for program files, multimedia files, text files, and documents, there are various communication protocols used, such as (1) hypertext transfer protocol (HTTP), which is designed and used for transferring a hypertext among two or more systems. HTML tags are used for creating links. These links may be in any form, such as text or images. HTTP is designed on client–server principles, which allow a client system for establishing a connection with the server machine for making a request. The server acknowledges the request initiated by the client and responds accordingly; (2) hypertext transfer protocol secure (HTTPS) is a standard protocol to secure the communi-cation between two computers, one using the browser and the other fetching data from the web server. HTTP is used for transferring data between the client browser (request) and web server (response) in the hypertext format, similar to the process of HTTPS except that the transferring of data is done in an encrypted format. Therefore, it can be inferred that HTTPS thwarts hackers from the interpretation or modification of data throughout the transfer of packets; (3) message query telemetry transport (MQTT), which is composed of MQTT client and broker (MQTT is based on TCP/IP protocol); and (4) transmission control protocol/internet protocol (TCP/IP), which is designed for the server and client network relation. Considering all the existing communication protocols, each has a distinct functionality that can be leveraged on a certain application.

In this study, the TCP/IP communication protocol is utilized over all the other com-munication protocols. The US Department of Defense created TCP/IP to define how com-puters communicate data from one device to another. TCP/IP places a high value on pre-cision, and it takes numerous steps to verify that data is delivered accurately between the two machines. With that, we consider using TCP/IP because we also only deploy the sys-tem between two devices: (1) an edge device and (2) a cloud device. TCP/IP breaks each message into packets, which are then reassembled at the other end. In fact, each packet could take a different route to the other computer if the first route is unavailable or congested. The TCP/IP communication protocol is divided into four layers, which ensures the accuracy of data processing. These layers include: (1) the datalink layer, (2) internet layer, (3) transport layer, and (4) application layer. These layers work together to process the data from one machine to another.

## 3. Proposed Edge–Cloud System

There has been a lot of debate on the strategies for the deployment of DL applications. One common way is to offload and perform tasks that are computationally expensive on the cloud [14,15]. The second approach is to perform computations closer to the edge and only send the metadata to the cloud [16,17]. Although the first approach fails to reduce latency, the second approach increases computational pressure on the edge device. In this study, we aim to reduce latency while maintaining the accuracy of DL applications. To achieve this, we implement a face recognition system on an edge–cloud network in which tasks are uniformly distributed between the edge and cloud device. The entire face recognition process is composed of two tasks: Face detection and face recognition.

As illustrated in Figure 1, the proposed edge–cloud network is made up of two sections: edge-based face detection and cloud-based face recognition. The task of face detection is carried out on the edge device in the first section, whereas the task of face recognition is carried out on the cloud in the second section. The two sections are integrated using a wireless communication protocol. The wireless communication protocol allows the transmission of video frames from the server to the client. In this study, the communication protocol utilized is the Transmission Control Protocol/Internet Protocol, also known as the TCP/IP communication protocol. The system’s primary objectives are to: (i) perform real-time face recognition, (ii) lower the computational burden on the edge device, and (iii) boost the system’s processing speed.

### 3.1. Edge-Based Face Detection

The edge-based face detection system consists of the (i) face detection algorithm and (ii) TCP/IP Server. The face detection task is employed in the edge device. This process starts with (1) initializing the camera for real-time video input. (2) The face detection system will cut the video feed into frames. (3) Utilizing the MTCNN function for face detection, the frames with detected faces will only be considered as an input for the face recognition function. (4) After processing the frame with a detected face, this detected face will be cropped from the input frame. After having a validated input for the face recognition system, the system will establish a connection with the cloud server by (1) specifying the IP address and port number of the server, (2) creating a client socket, (3) initiating the connection, (4) encoding the image into utf-8 encoding, and (5) sending the file and closing the socket connection.

#### 3.1.1. Face Detection Algorithm

The employed face detection algorithm is based on the multitask cascaded convolutional network (MTCNN). The MTCNN is a deep-learning-based approach for face and landmark detection having an accuracy of 96% and 92% for frontal face and side face, respectively, which is invariant to the head pose, occlusions, and illuminations. The locations of the face and landmarks are computed by a three-stage process. In the first stage, a fully convolutional network (FNN) called the proposal network (P-Net) is used to obtain the candidate windows and regression vectors. Subsequently, non-maximum suppression (NMS) is employed to merge highly overlapped candidates. In the second stage, all obtained candidates are fed into another CNN called the refine network (R-Net). This network rejects a huge number of false candidates and outputs whether the input is a face. In the third stage, the output network (O-Net) outputs five facial landmarks’ positions for eyes, nose, and mouth. Each stage involves three tasks: (1) face/non-face classification, (2) bounding box regression, and (3) facial landmark localization. The face/non-face classification challenge is a binary classification task. No face is classified when the output is zero, and a face is classified when the output is one. Bounding box regression is a type of regression task. This task determines the location of the facial bounding box. The challenge of locating facial landmarks is also a regression task. In this part, the positions of the face markers, such as the eyes, nose, and mouth, are further examined. This helps the system to identify the head pose more precisely.

#### 3.1.2. TCP/IP Server

As shown in Figure 2, the server end of the TCP/IP protocol works in conjunction with the MTCNN algorithm for face detection and transmission of frames containing the detected faces to the cloud. The server creates a socket object when it starts up. This means it associates a socket with the cloud device’s IP address and port number. The IP address is then bound to the socket. This is the same as giving the socket a name. The server attempts to connect to the client. It keeps track of any new connections that are made. It manages connections using the accept or close methods. The server establishes a connection with the client using the accept method and then uses the close method to terminate the connection. After the connection is established, the camera on the edge device is activated and the MTCNN face detection algorithm is used to check for faces. Subsequently, all cropped face images within the frames are sent to the cloud device’s client socket.

### 3.2. Cloud-Based Face Recognition

The cloud-based face recognition system is made up of the (i) face recognition algorithm and (ii) TCP/IP Client. After MTCNN has been applied in the edge device, a cropped image of only the detected face is extracted. The size of this image will be much smaller than the entire image’s size. This image is then received by the cloud for postprocessing with a face recognizer. Owing to the small size of the extracted face, the procedure will be significantly faster, and communication will use less bandwidth and energy.

#### 3.2.1. Face Recognition Algorithm

In this study, a Python face recognition module was utilized to implement face recognition. The library allows recognizing and manipulating faces using Python IDE or the command line interface (CLI). The library is built using Dlib’s state-of-the-art face recognition built with deep learning. The accuracy of the face recognition algorithm performs 99.38% on the labeled faces in the wild benchmark. This face recognition algorithm is deployed on the client side of the TCP/IP network. The client network must remember all the faces even if the machine is shut down and restarted, which is one of the challenges with the application. Consequently, we compiled a database of well-known individuals. Only faces that have been previously identified in the knowns database were detected when the face recognition algorithm was executed. The remaining faces were assigned as “unknown”. Consequently, a frame with a bonding box and a name label with the name of the recognized face were established.

The face recognition algorithm works by first finding the facial outline. The outline maps the facial features such as each person’s eyes, nose, mouth, and chin. Moreover, this face recognition library has a list of module content that can be used to further improve and manipulate the output. This library can be installed using Python language. Python 2.7 or Python 3.3+ can be used for the face recognition library dependency. With the facial images already extracted, cropped, and resized, the face recognition algorithm is responsible for finding the characteristics that best describe the image. A face recognition task is basically comparing the input facial image with all facial images from a database with the aim of finding the user that matches that face. It is basically a 1xN comparison. Once the face image is loaded, the face_encodings() function returns the 128-dimension embedding vector for each given face. This encoding process is performed using a ‘dlib_face_recognition_resnet_model_v1.dat’ model, which stores the recognized face image in a NumPy array. Thereafter, the face_distance() function gets a Euclidean distance for each comparison face. The Euclidean distances in the embedding space directly correspond to face similarity; faces of the same person have small distances and faces of distinct people have large distances. The matching name of the face which poses the smallest distance compared against the face encoding lists will be the final output.

#### 3.2.2. TCP/IP Client

On the cloud device, the TCP socket’s client end was implemented. When the client is initiated, it associates a socket with the IP address, as shown in Figure 3. The socket is then tied to the IP address. The client checks the TCP socket’s server for any connection requests. After establishing a connection, the frames containing the discovered faces are received for further face recognition. The output is the result of the entire face recognition process, indicating whether a face was recognized.

The necessity for a large computing capacity of a high-complexity, machine-learning algorithm is handled by partitioning the task across the edge and cloud networks. The processing becomes more efficient by running the face detection algorithm on the edge and the face recognition on the cloud. Face detection is implemented on the cloud in this article to address the latency issue, allowing the face detection algorithm to recognize faces and remove those frames before sending to the cloud.

## 4. Results and Discussions

In this study, a real-time face recognition inference application was deployed, using the MTCNN detector and Python’s face recognition library. The performance of the application was evaluated on (i) an edge device, (ii) a cloud device, and (iii) the proposed edge–cloud system. We provide a comparative analysis of the three deployments using input videos of varying resolutions and FPS values. Through a series of 45 experiments, the results demonstrate that the edge–cloud deployment is the most efficient implementation in terms of processing speed. We intend to extend this study to the use of deep learning frameworks for face recognition and the processing of live camera feed.

### 4.1. Experimental Setup

#### 4.1.1. Input Data

During the inferencing procedure, fifteen videos with varied duration, FPS values, and resolutions (HD and full HD) were used as input. The goal was to determine how varying resolutions and frames per second affect the processing speed of different deployment systems. Table 1 lists the information of these videos. Three sets of video durations (Video 1 = 50 s, Video 2 = 40 s, and Video 3 = 10 s) were used and tested as input. The durations of each video were classified into three FPS categories (30 FPS, 60 FPS, and 90 FPS). Moreover, the video dimensions of each video are also listed in Table 1.

#### 4.1.2. Edge Device

Originally, computing networks are classified into three main categories, namely: (1) edge, (2) fog, and (3) cloud. For this study, we only considered the edge and cloud network. Moreover, fog devices were also regarded as edge or cloud devices. The proposed edge–cloud system was implemented using an embedded device and a PC that serves as the cloud platform or device. The embedded device employed in this paper was the NVIDIA Jetson Nano. It is a single board computer that includes a 128-core Maxwell GPU and a quad-core ARM A57 64-bit CPU. For detecting faces, the prototype used a camera, the Raspberry Pi Camera version 2. It has a Sony IMX219 8-megapixel sensor, supports 1080p30, 720p60 and 640 × 480p90 video, and is connected via camera port using a short ribbon cable.

#### 4.1.3. Cloud Device

Instead of a cloud provider, in this study, a device serving as a cloud device was deployed. The cloud device is a desktop with Ubuntu 18.04.5 LTS. The desktop was equipped with an Intel i5-4690 CPU @ 3.50 GHz and a RAM size of 32 GB. To evaluate our face recognition system, we utilized random faces of known people saved in the cloud storage as a reference for face recognition.

### 4.2. Results Analysis

In each deployment, a pre-trained MTCNN face detection algorithm was used during the inferencing stage. The inference process was accelerated using the ONNX runtime.

Table 1 lists the FPS rates achieved for all deployments. From the results of each deployment, as expected, the edge shows a slow execution time. Jetson Nano clearly has limited computational capabilities when executing the MTCNN implementation. The cloud 1 deployment, running on Ubuntu 18.04.5, achieved a low execution time better than the edge deployment. The cloud 2 deployment, which receives the video stream, was faster than the former approach. The process can be performed by edge, cloud 1, or cloud 2 deployment. However, while running a high-complexity algorithm, the accuracy and processing speed are hampered because of the edge device’s limited computational capability.

The proposed edge–cloud deployment outperformed the edge deployment by 8.5 times and outperformed both cloud 1 and cloud deployments by 1.91 times This demonstrates how the cloud device’s computational capability can influence the performance of the deployed facial recognition model.

In Figure 4, a violin plot is employed because it is a method of plotting numeric data that can be considered as a combination of a box plot and a kernel density plot. Generally, the violin plot shows the same information as the box plot. The violin plot displays the (1) median, which is a white dot on the violin plot, (2) interquartile range, which is the black bar in the center of the violin, and (3) the lower/upper adjacent values, which are the black lines stretched from the bar. In this study, the violin plots show the density of the processing speed achieved by each of the deployments for the input video resolutions (HD and full HD). Two plots were constructed inside one figure by splitting the violin plots, which correspond to HD and FHD. Consequently, for every deployment, the HD videos are processed faster than the FHD videos.

The box plots in Figure 5 represent the density of the processing speed, which was achieved by all deployments, for the input video FPS (30, 60, and 90). As shown in Figure 5d, it can be inferred that for edge–cloud deployments, 30 FPS is the best input video FPS to use. In the edge and cloud 1 deployments in Figure 5a,b, video inputs with 60 FPS perform better compared to other deployments. Moreover, as depicted in Figure 5, video inputs with 90 FPS perform worse for all deployments. This poor performance can be attributed to the fact that all 90 FPS videos were FHD.

A better comparison is depicted in Figure 6, where the processing speed density of the HD and FHD videos for different FPS values is shown. For all deployments, videos with 90 FPS performed worse than other videos of the same FHD resolution. Figure 6a, Figure 6b, and Figure 6c show the result for the edge, cloud 1, and cloud 2, respectively. For the proposed edge–cloud system, 30 FPS video inputs achieved better performance compared to both 60 FPS and 90 FPS video inputs. Furthermore, when the proposed edge–cloud system is compared to other deployments, the proposed system achieved a higher FPS rate. This means that splitting the task of face detection and face recognition on the edge and cloud network is beneficial.

From these results, it can be deduced that the FPS and resolution of the input video play a key role in the processing speed of the deployment of face recognition applications. It is also clear that the best performance was achieved in the edge–cloud deployment with an input video of 30 FPS and HD resolution. Additionally, splitting the task of face detection and face recognition between the edge and cloud network shows an improvement in the processing speed and capability of the system.

## 5. Conclusions

In this study, a real-time face recognition inference application was deployed, using the MTCNN detector and Python’s face recognition library. The performance of the application was evaluated on (i) an edge device, (ii) a cloud device, and (iii) the proposed edge–cloud system. We provided a comparative analysis of the three deployments using input videos of varying resolutions and FPS values. Through a series of 45 experiments, the results demonstrated that the edge–cloud deployment is the most efficient implementation in terms of processing speed. We intend to extend this study to the use of deep learning frameworks for face recognition and the processing of live camera feed.

## Figures and Tables

**Figure 1 sensors-22-08371-f001:**
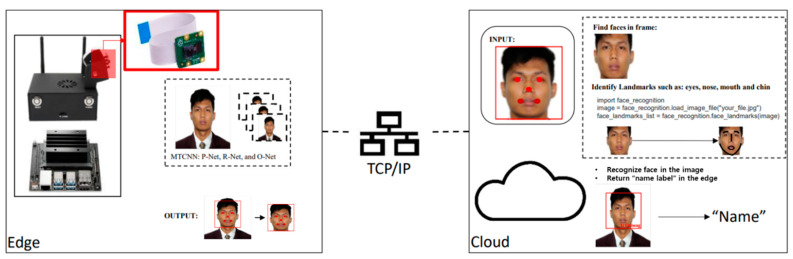
Proposed edge–cloud system.

**Figure 2 sensors-22-08371-f002:**
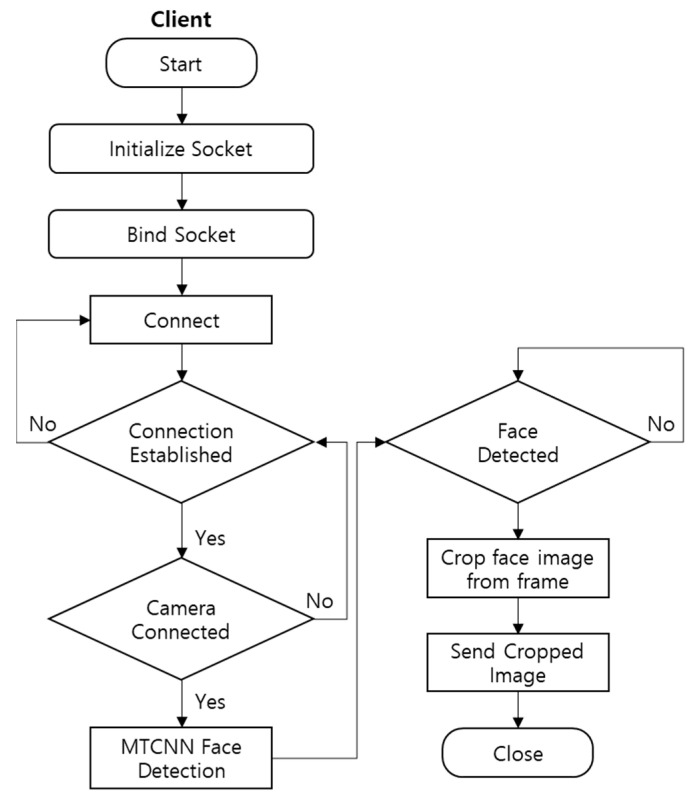
Edge-based face detection.

**Figure 3 sensors-22-08371-f003:**
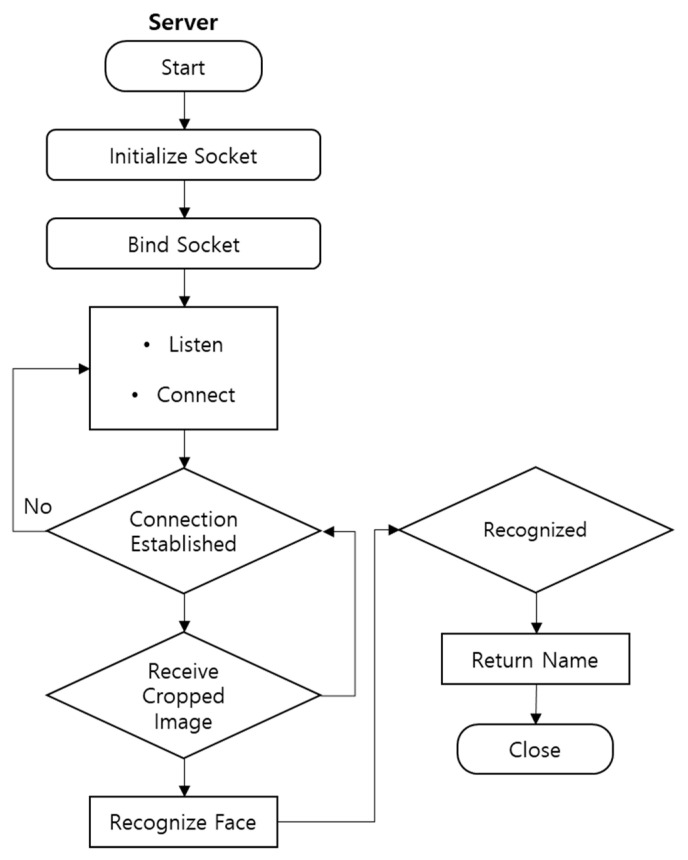
Cloud-based face recognition.

**Figure 4 sensors-22-08371-f004:**
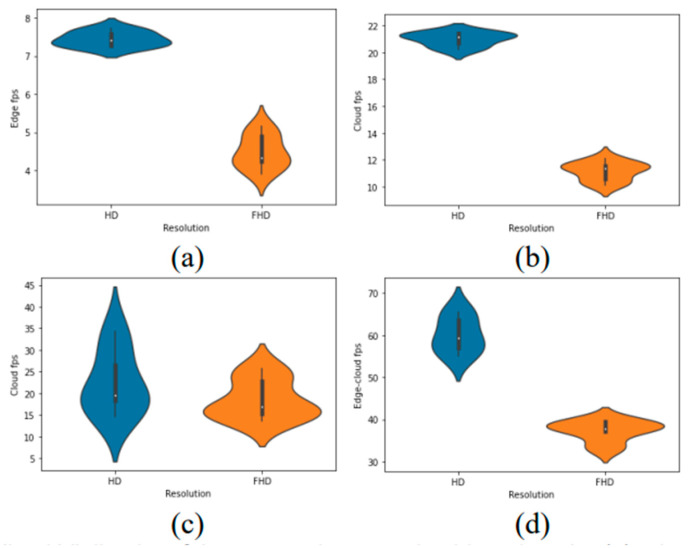
Violin plot of the processing speed achieved on the (**a**) edge, (**b**) cloud 1, (**c**) cloud 2, and (**d**) edge–cloud for each input video resolution.

**Figure 5 sensors-22-08371-f005:**
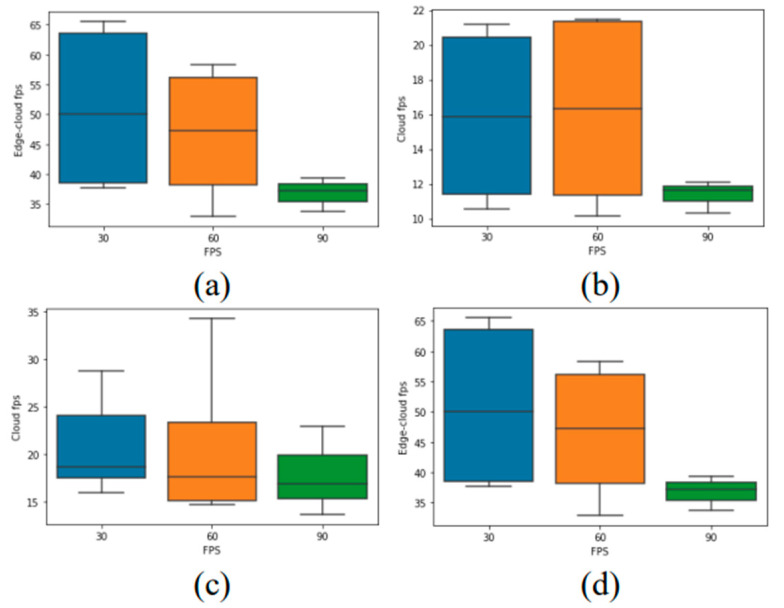
Box plot of the processing speed achieved on the (**a**) edge, (**b**) cloud 1, (**c**) cloud 2, and (**d**) edge–cloud for each input video FPS.

**Figure 6 sensors-22-08371-f006:**
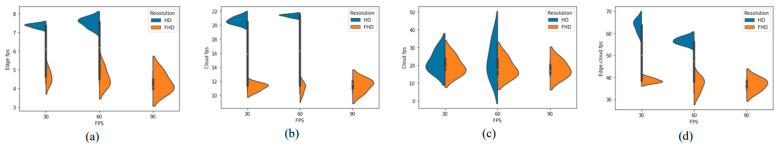
Violin plot of the processing speed achieved on the (**a**) edge, (**b**) cloud 1, (**c**) cloud 2, and (**d**) edge-cloud for each input video FPS and resolution.

**Table 1 sensors-22-08371-t001:** Results for each deployment for the fifteen input videos.

Video	Duration	FPS ^a^	Resolution	Width	Height	Edge FPS [15]	Cloud 1 FPS [14]	Cloud 2 FPS [14]	Edge-Cloud FPS
Video 1	50	30	HD	1280	720	7.40	20.56	28.72	60.34
			FHD	1920	1080	4.26	10.56	25.67	39.76
		60	HD	1280	720	7.24	21.12	34.27	56.52
			FHD	1920	1080	4.24	10.16	24.52	32.95
		90	FHD	1920	1080	3.90	10.31	22.93	39.39
Video 2	40	30	HD	1280	720	7.14	20.21	19.27	65.51
			FHD	1920	1080	4.54	11.54	15.92	38.23
		60	HD	1280	720	7.72	21.47	19.87	55.11
			FHD	1920	1080	4.33	11.58	05.03	37.77
		90	FHD	1920	1080	4.08	12.12	13.03	33.77
Video 3	10	30	HD	1280	720	7.44	21.19	18.07	64.65
			FHD	1920	1080	4.98	11.37	17.34	37.76
		60	HD	1280	720	7.63	21.46	14.67	58.23
			FHD	1920	1080	5.15	11.29	15.31	39.39
		90	FHD	1920	1080	4.89	11.67	16.89	37.19

## Data Availability

Not applicable.

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
