# Peer review of "Optimizing Face Recognition Inference with a Collaborative Edge–Cloud Network"

_sensors, 2022, doi:10.3390/s22218371_

Round 1
Reviewer 1 Report
11 The subject matter is interesting but the authors recall well-known tools and protocols.
22. What is the novelty and originality of this work?
3 3. In the context of biometrics, and more precisely of facial recognition and edge-cloud network use, the authors should analyze and cite the following key references:
Ear Recognition Based on Deep Unsupervised Active Learning IEEE Sensors Journal, vol. 21, no. 18, pp. 20704–20713, 15 Sept.15, 2021, doi: 10.1109/JSEN.2021.3100151
Past, present, and future of face recognition: A review, Electronics, vol. 9, no. 8, 1188, 2020, doi: 10.3390/electronics9081188.
3D Face Recognition Using Multiview Keypoint Matching, 2009 Sixth IEEE International Conference on Advanced Video and Signal Based Surveillance, 2009, pp. 290-295, doi: 10.1109/AVSS.2009.11.
Cloud-Vision: Real-time face recognition using a mobile-cloudlet-cloud acceleration architecture, 2012 IEEE Symposium on Computers and Communications (ISCC), 2012, pp. 000059-000066, doi: 10.1109/ISCC.2012.6249269.
Deep Unified Model For Face Recognition Based on Convolution Neural Network and Edge Computing, in IEEE Access, vol. 7, pp. 72622-72633, 2019, doi: 10.1109/ACCESS.2019.2918275
4. This work needs to be more thorough and requires quantitative measures and other metrics.
Reviewer 2 Report
See attached file.

Reviewer 3 Report
In this work, authors have proposed a real -time face recognition inference application which is deployed, using
the MTCNN detector and python’s face recognition library. Further, the performance of the application was evaluated on an edge device, a cloud device, and the proposed 378
edge-cloud system. The work is intereseted however need substaintial revisions before publication as given below.
1. What is novelity of the work.Please underscore the scientific value added/contributions of your paper in your abstract and introduction and address your debate shortly in the abstract.
2. A good article should include, (1)originality, new perspectives or insights; (2) international interest; and (3) relevance for governance, policy or practical perspectives relevant to the focus of this manuscript. Please emphasis the originality of your work.
3. How edge infrasturure is designe. Provide details.
4. What is the accuracy of face detection method? why it is not discuused in the manuscript. mention it.
5. Python 2.7 or Python 3.3+ can be used for the face recognition library dependency. But need to discuss algorithm utilized in library.
6. Can you implement edge comuting at algorithm level.
7. which dataset is utilized for evaluation.
8. Literature review need to be enhanced.
Round 2
Reviewer 1 Report
The authors took into account our recommendations and answered our questions. The manuscript has been considerably improved and deserves to be published.
Reviewer 2 Report
All of my comments were properly addressed. Well done!
Reviewer 3 Report
Authors have addressed all the major concerns/ revisions in the revised manuscript. Manuscript is acceptable in its present form.